# ORBIS: OPEN DATASET CAN RESCUE YOU FROM DATASET BIAS

## ABSTRACT

Dataset bias, in the context of machine learning, pertains to the issue of unintended correlations between target labels and undesirable features found in specific training datasets. This phenomenon frequently arises in real-world scenarios and can lead to unintended behaviors. Researchers have devised techniques to alleviate this bias by diminishing the influence of samples with spurious correlations (*i.e.,* bias-aligned samples) while assigning greater importance to other samples (*i.e.,* bias-conflicting samples) during the training process. Prior approaches have mainly focused on leveraging given training datasets and have not explored the potential of harnessing open datasets, which have huge size of samples. Nonetheless, open datasets may contain noisy information posing a challenge for straightforward integration. In this paper, we introduce a novel method calld `ORBIS` to tackle dataset bias using open datasets. `ORBIS` comprises two core components. Firstly, it involves the selection of relevant samples from open datasets whose context aligns with the characteristics of the given training dataset. Subsequently, a debiased model is trained using both training dataset and selected samples. We assess the effectiveness of this proposed algorithm in conjunction with established debiasing methods and evaluate its performance on both synthetic and real-world benchmark datasets.

## 1 INTRODUCTION

Deep neural networks (DNNs) have demonstrated remarkable capabilities across various applications, such as classification He et al. (2016); Dosovitskiy et al. (2020), object detection Girshick (2015), and image generation Goodfellow et al. (2020). While these networks often achieve impressive performance by leveraging well-curated and accurately labeled datasets, research has shown that their generalizability can deteriorate when confronted with spurious correlations between intended and unintended features. This phenomenon is commonly referred to as the *dataset bias problem*. In real-world dataset collection, spurious correlations frequently emerge. For instance, when gathering a dataset for classifying "ski" images, a significant portion of the images might be captured against a snowy background. In this case, the images of "ski" exhibit a spurious correlation with "snow," and the snowy background is considered the *bias attribute*. These images are referred to as *bias-aligned samples*. Conversely, samples captured in a desert environment (e.g., desert skiing) would be *bias-conflicting* samples. When training a model on this dataset, the model suffers from the dataset bias problem, as it becomes more influenced by the "background" (snow) rather than the "ski" itself.

In prior research, several strategies have been advanced to tackle the dataset bias issue by incorporating human-supervised annotations related to bias. For instance, the authors of Arjovsky et al. (2019); Sagawa et al. (2019); Li & Vasconcelos (2019); Kim et al. (2019); Tartaglione et al. (2021) utilized bias-related labels to prevent the trained models from learning these annotations. In Kim et al. (2019), adverasarial training was employed to make the trained model fail to learn the bias attribute. While these labeling-intensive methods have proven effective, they come at a significant cost in terms of human effort. To alleviate this burden, researchers have made assumptions about existing bias in datasets and utilized these assumptions in their methodologies Lee et al. (2019); Bahng et al. (2020b); Alvi et al. (2018). For instance, in Alvi et al. (2018), the authors transformed the image into grayscale to prevent color bias. Nonetheless, even these approaches require some level of human supervision and may not be readily applicable in scenarios where the biased feature is not explicitly accounted for, such as object bias.

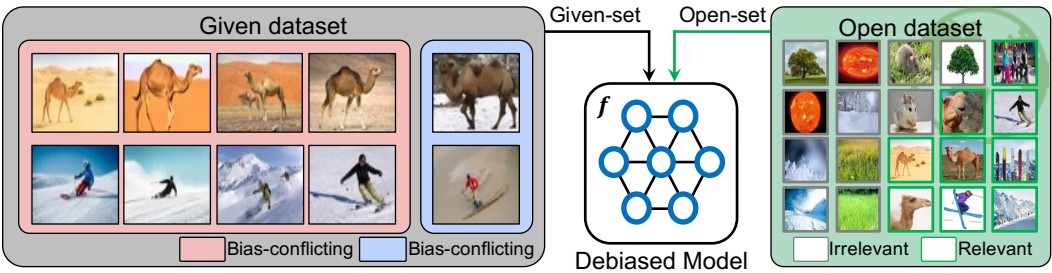

Figure 1: Despite prior methods primarily focusing on mitigating dataset bias using only the provided training dataset, this paper aims to tackle the problem by incorporating both the training dataset and an open dataset that is available but presents challenges in extracting relevant samples.

To minimize human involvement, researchers have explored approaches to replace human intervention with a biased model ($f_b$) and provide guidance on which samples have to emphasize or underweight in order to achieve balance. Notable studies in this area include Ahn et al. (2023); Lee et al. (2021); Kim et al. (2021); Le Bras et al. (2020); Nam et al. (2020); Kim et al. (2022); Hwang et al. (2022). For instance, Nam et al. (2020); Lee et al. (2021) employed relative difficulty weights, proportional to the per-sample loss obtained from the biased model, to train the debiased model ($f_d$) by assigning weights to each sample. Recently, Hwang et al. (2022) proposed an effective mixup algorithm for debiasing that combines bias-conflicting and bias-aligned samples, while Kim et al. (2022); Le Bras et al. (2020) leveraged ensemble networks to identify samples that require emphasis. In a different research direction, Ahn et al. (2023) introduced a two-stage debiasing approach that utilizes the per-sample gradient norm and adjusts the sampling probability of each sample based on the obtained value. Two key insights emerge from these efforts: the identification of bias-conflicting samples and the amplification of their importance. This leads to the intuitive understanding that accentuating the role of bias-conflicting samples can help in alleviating biased outcomes.

In the stream of the research that utilizes the given dataset, however, we have a following question: *Can open-world datasets, which are uncurated but composed of a huge amount of data, help mitigate the dataset bias problem?* We mainly try to answer the question by constructing a module that helps debiasing algorithms. To this aim, we study following questions. (1) What is the key factor to improve the performance of mitigating dataset bias. (2) How can we effectively utilize open datasets to address the dataset bias problem? Based on these questions, we propose a method that automatically mines samples from the open dataset and leverages them to mitigate the dataset bias problem.

**Contribution.** Based on the above questions, we briefly summarize our contributions as follows:

- We first investigate which one plays a more crucial role in mitigating dataset bias: the number of the proportion of bias-conflicting samples. In short, we find that the number is the dominant factor. Therefore, we need to obtain additional bias-conflicting samples from an external data source.

- To the bset of our knowledge, we explore ways to leverage open datasets to increase the number of bias-conflicting samples. (The problem formulation is described in Figure 1) However, leveraging open datasets for addressing dataset bias is challenging for tow reasons: (1) Open datasets primarily consists of irrelevant samples for the given task. (2) Even if we have access to relevant samples, the labels of these samples are either absent or mislabeled.

- We propose a novel algorithm, called ORBIS, which selects relevant datasets from the open dataset and effectively leverages them to tackle the dataset bias problem. The proposed algorithm consists of two main steps: (1) Extracting relevant samples from a large open dataset, which may contain highly correlated or uncorrelated samples. (2) Implementing a debiasing algorithm using a constrastive loss that does not require per-sample labels.

- Building upon the proposed method, we investigate whether properly leveraging open datasets can enhance debiasing performance. We apply the proposed method to various benchmark datasets including bFFHQ Lee et al. (2022); Kim et al. (2021; 2022), Dogs and Cats Lee et al. (2022), and CelebA Nam et al. (2020).

## 2 Dataset Bias Problem and Open Dataset

In this section, we investigate the phenomenon of dataset bias, particularly in the context of open datasets. First, we elucidate our problem setting, complete with pertinent notations, to facilitate a clear understanding of our methodology.

### 2.1 Dataset Bias Problem

**Dataset bias problem.** We provide a precise explanation of the notations and issues related to the dataset bias problem. We assume that we have a training dataset denoted as $\mathcal{D}_{\text{tr}} = \{(x_i, y_i)\}_{i=1}^{N}$, where $(x_i, y_i) \sim \mathcal{X} \times \mathcal{Y}$. Each data point consists of an input value $x_i$ and its corresponding label $y_i$, which belongs to one of $C$ classes. The dataset comprises $N$ total data points, each of which has features like camel/ski and background, as exemplified in Figure 1. Our primary objective is to achieve accurate calssification on the Ski/Camel attribute (referred to as the *target attribute*) while mitigating the influence of the unintended attribute (referred to as the *bias attribute*). We focus on instances where a significant majority of data points are strongly correlated with both the target and bias attributes; these are termed as *bias-aligned samples*. In contrast, a limited set of data points exhibit weak correlation, which we denote as *bias-conflicting samples*. We introduce the metric $\rho$ to quantitatively measure the proportion of bias-conflicting samples, referred to as the *bias-conflicting ratio*.

If the model finds it easier to learn the bias attribute (e.g., Ski/Camel) rather than the target attribute (e.g., Background), it may fail to learn the target attribute accurately. This observation has been highlighted in previous work Nam et al. (2020). Consequently, during the inference phase, the performance of the trained model may be poor, resulting in the undesirable situation known as the dataset bias problem. Therefore, guiding correct direction to answer is important.

**Summary of prior debiasing approaches.** Previously, mainly two networks are used, as briefly summarized in section 1, when bias-related information is not provided. Initially, the biased model $f_b$ is trained on the given dataset $\mathcal{D}_{\text{tr}}$. Here, a technique for emphasizing bias-aligned samples is leveraged to enhance the distinguishability of bias-conflicting samples. Based on information from the biased model $f_b$, the degree of emphasis for each sample is determined using several metrics, such as training loss Nam et al. (2020); Lee et al. (2021), gradient-norm Ahn et al. (2023), softmax entropy Ahn & Yun (2023), and per-sample accuracy Kim et al. (2022). In this step, bias-conflicting samples receive a higher degree of emphasis compared to bias-aligned samples, and as a result, the debiased model $f_d$ is trained in a balanced manner.

### 2.2 Open dataset

In practical scenarios, we encounter a myriad of open datasets, described by a distribution $\mathcal{D}_{\text{open}} = \{x_j\}_{j=1}^{M} \sim \mathcal{X}_{\text{open}}$. This distribution may differ from the target distribution $\mathcal{X}$. Importantly, while some open datasets may carry their own class labels, relying solely on this information can be problematic, and in certain scenarios, the labels may even be absent. Our investigation thus concentrates on situations where the open dataset does not offer specific information concerning either class categories or per-sample labeling.

## 3 Motive Observation

| Alg. | w/o $\mathcal{D}_{\text{rel}}$ | w/ $\mathcal{D}_{\text{rel}}$ | Δ |
|---|---|---|---|
| Vanilla | 55.24 | 57.88 | +2.64 |
| LfF | 63.34 | 68.23 | +4.89 |
| LfF + BE | 65.14 | 71.26 | +6.12 |
| Disent | 61.21 | 66.40 | +6.19 |
| Disent +BE | 65.36 | 70.43 | +5.07 |

| | $\rho = 1\%$ | | | Conflicting # = 192 | | |
|---|---|---|---|---|---|---|
| Algn. # / Conf. # | **19,008/192** | 15,206/153 | 9,504/96 | 17,180/192 | 13,362/192 | 9,544/192 |
| Conf. Acc. (Vanilla) | **60.96** | 59.42 | 52.13 | 60.67 | 61.03 | 60.83 |
| Conf. Acc. (LfF) | **69.24** | 65.43 | 59.81 | 68.62 | 68.76 | 68.92 |
| | $\rho = 2\%$ | | | Conflicting # = 384 | | |
| Algn. # / Conf. # | 18,816/384 | 15,052/307 | **9,408/192** | 16,934/384 | 13,171/384 | 9,408/384 |
| Conf. Acc. (Vanilla) | 69.00 | 65.65 | **61.52** | 68.92 | 67.92 | 68.72 |
| Conf. Acc. (LfF) | 73.08 | 72.24 | **68.93** | 73.24 | 72.16 | 73.28 |

Table 1: Relevant or Irrelevant                    Table 2: Portion v.s. Number

In this section, we scrutinize three key questions: (i) Which is more significant—the number or the ratio of bias-conflicting samples? (ii) How crucial is it to acquire samples relevant to $\mathcal{D}_{\text{tr}}$ from $\mathcal{D}_{\text{open}}$.

(iii) What impact does the label of the open dataset have? To investigate these questions, we evaluate the bFFHQ Kim et al. (2021) with the WebVision open dataset Li et al. (2017).

**Observation 1: Extracting target task-relevant samples is crucial.** To assess the influence of these relevant samples, we manually curate a set of 150 relevant samples for each class from the open dataset and assign label to them. Details regarding the manual selected WebVision samples are provided in Appendix A. For comparison, we also choose random samples and assign labels to them arbitrarily. As indicate in Table 1, the selection of relevant samples significantly outperforms the arbitrary selection. This underlines the necessity of extracting or selecting samples related to the target dataset from the open dataset to improve debiasing performance.

**Observation 2: Quantity is more crucial than the ratio of bias-conflicting samples.** We explore two scenarios to identify the leading factor that influences debiasing: (1) maintaining the ratio of bias-conflicting samples in the training dataset and comparing the accuracy of these samples, and (2) maintaining the number of bias-conflicting samples in the training dataset and evaluating their accuracy. As detailed in Table 2, even though the bias-conflicting ratio is slightly higher, the accuracy of bias-conflicting sampels remains relatively similar when the number of bias-conflicting samples is kept constant. Furthermore, when bias-conflicting samples are maintained, meaning the ratio is not identical, the accuracy remains similar. In summary, it is imperative to acquire additional bias-conflicting samples unless the bias-conflicting ratio ($\rho$) decreases.

**Observation 3: Label condition is of paramount importance.** In our final discovery, we examine the labeling condition after obtaining the relevant samples, a necessary step because open datasets often feature noisy or unlabeled dataset Li et al. (2017). We investigate whether pseudo-labeling, which assigns labels based on cosine similarity trained on the WebVision dataset using SimCLR training procedure, and FixMatch Sohn et al. (2020), a semi-supervised learning method, prove effective. As

Table 3: Pseudo-labeling results.

| Algorithm | $\rho = 1\%$ | $\rho = 2\%$ |
|---|---|---|
| Vanilla | 60.96 | 69.00 |
| + Pseudo-Label | 56.23 | 61.75 |
| + FixMatch | 61.03 | 69.22 |
| LfF | 69.24 | 73.08 |
| + Pseudo-Label | 58.43 | 62.42 |
| + FixMatch | 69.67 | 73.68 |

indicated in Table 3, semi-supervised debiasing is not a straightforward process and can potentially degrade performance. This decrement mainly arises because high-confidence bias-aligned samples are often selected for pseudo-labeling (examples are described in Appendix F). Therefore, crafting a suitable labeling strategy for these relevant samples is paramount.

**Key motivation for consturcting an algorithm utilizing an open dataset.** Based on the findings above, several essential components are necessary when leveraging open datasets to address the bias problem. (1) Avoid focusing solely on selecting bias conflicting samples: it is not necessary to exclusively focus on selecting bias-conflicting samples in order to increase the ratio of bias-conflicting instances to improve the accuracy of bias-conflicting samples. (2) Select relevant samples from the open dataset: it is imperative to select relevant samples from the open dataset, ensuring they are aligned with the target task. (3) Consider the implictions of pseudo-labeling: be cautious when employing pseudo-labeling, as it has the potential to degrade debiasing performance Ahn & Yun (2023). This is because pseudo-labelign relies on confidence-based labeling, which may not align with the debiasing objectives. By taking these considerations into account, the construction of an algorithm utilizing open dataset can be more effective in addressing dataset bias challenges.

## 4 ORBIS: OPEN DATASET CAN RESCUE YOU FROM BIAS

In this section, we present a comprehensive description of the algorithm. Proposed algorithm consists of two primary steps: (1) selecting a relevant samples, and (2) training debiased model. In the initial step, our objective is to choose a subset of the open dataset, denoted as $\mathcal{D}_{rel} \subset \mathcal{D}_{open}$, that closely resembles the given training dataset $\mathcal{D}_{tr}$. The purpose of this step is to gather relevant samples, as indicated in **Observation 1**, and ensure that we have an adequate number of bias-conflicting samples as highlighted in **Observation 2** in section 3. In the subsequent step, we train a debiased model that will be applied during the testing phase. However, this task is complicated by the presence of unreliable labels in the dataset. To tackle this issue, we emply a self-supervised learning mechanism that does not rely on target labels. We will now proceed to explain how we identify the relevant dataset and train the debiased model using the selected subset. An overview of the proposed algorithm is provided in Figure 2 and pseudo-code in Appendix B.

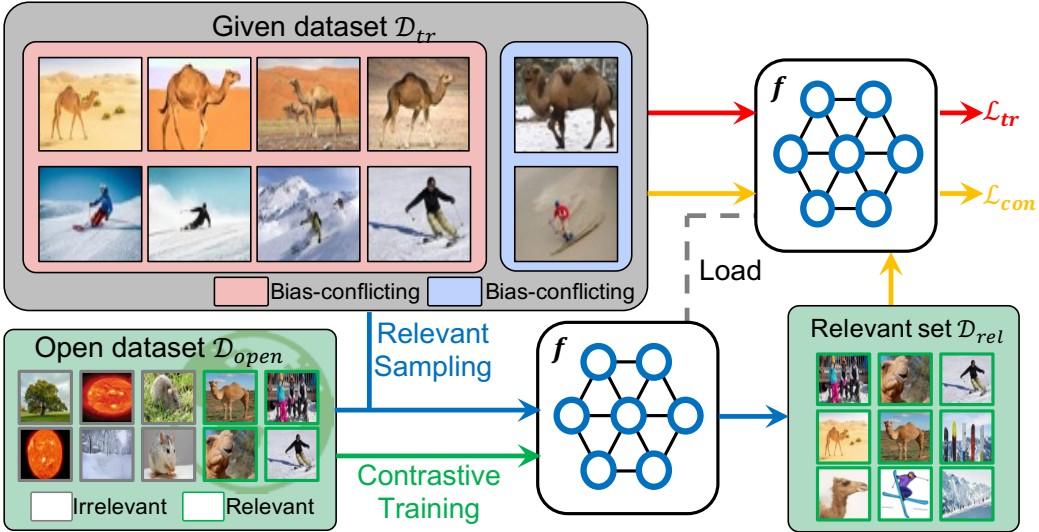

Figure 2: Overview of the proposed algorithm. ORBIS composed of three steps: (1) Unsupervised training on the open dataset $\mathcal{D}_{\text{open}}$. (2) Extract relevant samples from open dataset. (3) Debiasing training and utilizing relevant dataset for final training.

**Cosine Similarity.** We utilize the widely-adopted metric for measuring the distance between two vectors, $z_i$ and $z_j$, known as cosine similarity. Mathematically, it is expressed as $\text{sim}(z_i, z_j) = \frac{z_i \cdot z_j}{|z_i| \cdot |z_j|}$. This cosine similarity severs two main purpose in this paper: (1) As a distance measure for subsampling the open dataset in Step 1. (2) For computing the contrastive loss in Step 2. Apart from the notations introduced above, which are specifically defined within each respective step, the following subsections will provide further explanations and details of our approach.

## 4.1 STEP 1: SELECTING THE RELEVANT DATASET

As previously noted, the open dataset $\mathcal{D}_{\text{open}}$ does not inherently contain samples that are directly relevant to the target task. Therefore, it is important to acquire samples that are specifically aligned with the target task. For example, if the training dataset $\mathcal{D}_{\text{tr}}$ comprises facial images, then the relevant dataset $\mathcal{D}_{\text{rel}}$ should encompass facial images extracted from $\mathcal{D}_{\text{open}}$. To achieve this objective, we initiate the training of a model using $\mathcal{D}_{\text{open}}$. Given that $\mathcal{D}_{\text{open}}$ can either be labeled with noisy anootations or remain unlabeled, we can effectively emply self-supervised learning (SSL) techniques that do note neccessitate labels. In particular, we train the model $f(x)$ based on the SimCLR framework Chen et al. (2020a)[1].

Following the training of the self-supervised model $f(x)$, our next step involves selecting samples from the open dataset that exhibit features similar to the class-wise centroids of the provided target dataset. The class-wise centroid of the target dataset is defined as follows:

$$\mathsf{C}_c = \frac{\sum_{(x_i, y_i) \in \mathcal{D}_{\text{tr}}^c} f(x_i)}{|\mathcal{D}_{\text{tr}}^c|} \quad \text{where } \mathcal{D}_{\text{tr}}^c = \{(x_i, y_i) | y_i = c \text{ and } (x_i, y_i) \in \mathcal{D}_{\text{tr}}\}, \quad (1)$$

where $\mathcal{D}_{\text{tr}}^c$ represents the set of samples in the target dataset belonging to class $c$, and $|\mathcal{D}_{\text{tr}}^c|$ denotes the cadinalirty of the set. To assess the distance between the class-wise centroid $\mathsf{C}_c$ and each sample in $\mathcal{D}_{\text{open}}$, we calculate the cosine similarity between $\mathsf{C}_c$ and $f(x_j)$, where $\bar{x}_j \in \mathcal{D}_{\text{open}}$ as follows:

$$\text{sim}(\mathsf{C}_c, f(x_j)) = \frac{\mathsf{C}_c \cdot f(x_j)}{\|\mathsf{C}_c\| \cdot \|f(x_j)\|} \quad (2)$$

After calculating the cosine similarity scores for each sample in $\mathcal{D}_{\text{open}}$, we generate a matrix of size $C \times |\mathcal{D}_{\text{open}}|$, where each entry represents the similairty score between a class-wise centroid and an

---

[1]For simplicity, we omit the projection head $h(\cdot)$ notation.

open dataset sample. From these similarity score, we identify the class-wise relevant set $\mathcal{D}_{\text{rel}}^c$ for each class ,which comprises samples exhibiting higher similarity. We determine membership in this set using the following condition:

$$\mathcal{D}_{\text{rel}}^c = \{x_j | \text{sim}(\mathtt{C}_c, f(x_j) \geq \tau, x_j \in \mathcal{D}_{\text{open}}\}, \tag{3}$$

where $\tau$ is the threshold hyperparameter. We construct the overall relevant dataset $\mathcal{D}_{\text{rel}}$ by taking the union of the class-wise relevant sets:

$$\mathcal{D}_{\text{rel}} = \bigcup_{c=1}^{C} \mathcal{D}_{\text{rel}}^c. \tag{4}$$

The approach of sampling the open dataset based on similarity was originally introduced in the field of self-supervised learning Kim et al. (2023a). The key difference between their work and our approach lies in how we employ the open dataset to train the $f(x)$ model. We choose to utilize the open dataset for training $f(x)$ because it enables the model to acquire expertise in handling specific characteristics of the open dataset, thereby ensuring that irrelevant samples are not included in $\mathcal{D}$rel. Furthermore, training $f$open on the open dataset offers the advantage of reducing computational costs, as the same $f_{\text{open}}$ model can be applied across various datasets without the need for retraining in each case.

## 4.2 STEP 2: TRAIN DEBIASED MODEL BY LEVERAGING RELEVANT SET

Once we have obtained $\mathcal{D}_{\text{rel}}$, which may contain missing or incorrectly labeled samples from the target task perspective, we design an algorithm to utilize it conservatively without relying on provided or pseudo-labels. In essence, we leverage contrastive loss for these samples. At this point, we have two datasets that can be used for training $f$: $\mathcal{D}_{\text{tr}}$ and $\mathcal{D}_{\text{rel}}$. Both datasets can be employed for training, especially based on contrastive loss. To incorporate both datasets, we merge them into a unified dataset as follows:

$$\mathcal{D}_{\text{rel+tr}} = \mathcal{D}_{\text{rel}} \cup \mathcal{D}_{\text{tr}}' \quad \text{where } \mathcal{D}_{\text{tr}}' = \{x | (x, y) \in \mathcal{D}_{\text{tr}}\}.$$

**Contrastive Loss.** Before delving into the details of contrastive loss, let us introduce some relevant notations. First, we consider the loss based on a randomly selected mini-batch of size $B$ from $\mathcal{D}_{\text{rel+tr}}$. Adhering to the principles of contrastive loss, we create two randomly augmented images, $(x_{2b}, x_{2b-1}) = (\mathcal{A}(x_b), \mathcal{A}(x_b))$, where $\mathcal{A}(x)$ denotes the augmentation operation and b is the index within the mini batch. When we obtain features $z$ from the projection head $z = h(f(x))$ the contrastive loss is expressed as follows:

$$\mathcal{L}_{\text{con}} = \frac{1}{2B} \sum_{b=1}^{2B} [\ell_{2b-1,2b} + \ell_{2b,2b-1}], \text{ where } \ell_{i,j} = -\log \frac{\exp(\text{sim}(z_i, z_j)/\kappa)}{\sum_{b=1}^{2B} \mathbb{1}_{b \neq i} \exp(\text{sim}(z_i, z_b)/\kappa)}. \tag{5}$$

Here, $\mathbb{1}_{b \neq i}$ represents the indicator function, and $\kappa$ is a temperature hyperparameter.

**Ultimate training.** Using the $\mathcal{L}_{\text{tr}}$ and $\mathcal{L}_{\text{con}}$ losses, we train the final target model $f$ for a total of $E$ epochs. The loss function for this training is defined as follows

$$\mathcal{L} = \mathcal{L}_{\text{tr}} + \lambda \mathcal{L}_{\text{con}}, \tag{6}$$

where $\lambda$ represents the balancing hyperparameter. Here $\mathcal{L}_{\text{tr}}$ is the debiasing loss, which can be a weighted cross-entropy loss. For simplicity, we set $\lambda = 0.01$ for all our experiments. It is worth noting that the mini-batches used for $\mathcal{L}_{\text{tr}}$ and $\mathcal{L}_{\text{con}}$ are not identical, as $\mathcal{D}_{\text{tr}}$ and $\mathcal{D}_{\text{rel+tr}}$ are different dataset. Therefore, we construct two types of mini-batches for each iteration and compute losses seprately.

## 5 EXPERIMENT

In this section, we will outline the experimental setup used to evaluate the performance of the proposed algorithm. We will provide comprehensive details regarding the biased datasets utilized in our experiments and describe the implementation of our algorithm. Following that, we will conduct a comparative analysis of the performance of ORBIS in comparison to previous algorithms.

Table 4: Unbiased test accuracy on bFFHQ, BAR and Dogs & Cats dataset. We format the best results as **bold**. We report the average results of three random trials. We remark the cases, e.g., % denotes the bias-conflicting ratio. L and T represents if the algorithm requires bias-related labels or bias-type. ✓ denotes it requires label or type information and ✗ represents that it does not.

| Algorithm | L/T | bFFHQ | | | | Dogs & Cats | | BAR | |
|---|---|---|---|---|---|---|---|---|---|
| | | 0.5% | 1% | 2% | 5% | 1% | 5% | 1% | 5% |
| w/o SimCLR model (ResNet-18) | | | | | | | | | |
| Vanilla | ✗/✗ | 55.64 | 60.96 | 69.00 | 82.88 | 48.06 | 69.88 | 70.55 | 82.53 |
| + ORBIS | ✗/✗ | 59.23 | 64.43 | 73.18 | 84.32 | 45.60 | 72.10 | 70.92 | 82.85 |
| HED | ✗/✓ | 56.96 | 62.32 | 77.72 | 83.40 | 46.76 | 72.60 | 70.48 | 81.20 |
| LNL | ✓/✓ | 56.88 | 62.64 | 69.80 | 83.08 | 50.90 | 73.96 | - | - |
| EnD | ✓/✓ | 55.96 | 60.88 | 69.72 | 82.88 | 48.56 | 68.24 | - | - |
| ReBias | ✗/✓ | 55.76 | 60.68 | 69.60 | 82.64 | 48.70 | 65.74 | 73.04 | 83.90 |
| LfF | ✗/✗ | 65.19 | 69.24 | 73.08 | 79.80 | 71.72 | 84.32 | 70.16 | 82.95 |
| + BE | ✗/✗ | 67.36 | 75.08 | 80.32 | 85.48 | 81.52 | 88.60 | 70.33 | 83.13 |
| + ORBIS | ✗/✗ | 68.62 | 75.63 | 76.41 | 81.31 | 74.51 | 86.53 | 72.41 | 83.52 |
| + BE + ORBIS | ✗/✗ | **69.28** | **78.61** | **85.43** | **89.42** | **81.96** | 89.42 | **74.52** | 84.12 |
| Disent | ✗/✗ | 62.08 | 66.00 | 69.92 | 80.68 | 65.74 | 81.58 | 70.33 | 83.13 |
| + BE | ✗/✗ | 67.56 | 73.48 | 79.48 | 84.84 | 80.74 | 86.84 | 73.29 | 84.96 |
| + ORBIS | ✗/✗ | 66.64 | 69.66 | 74.86 | 84.21 | 76.50 | 88.30 | 72.49 | 83.33 |
| + BE + ORBIS | ✗/✗ | 68.54 | 75.60 | 80.88 | 86.23 | 81.81 | **89.62** | 74.31 | **85.23** |
| w/ SimCLR model (ResNet-50) | | | | | | | | | |
| Vanilla | ✗/✗ | 56.21 | 69.23 | 74.12 | 82.21 | 66.54 | 86.43 | 49.21 | 60.81 |
| + ORBIS | ✗/✗ | 61.83 | 70.21 | 76.31 | 87.43 | 73.32 | 89.43 | 50.31 | 61.69 |
| LfF | ✗/✗ | 65.62 | 74.52 | 77.83 | 84.31 | 72.91 | 90.83 | 55.01 | 66.26 |
| + ORBIS | ✗/✗ | **71.63** | **80.31** | 83.21 | **88.38** | 76.32 | 91.23 | **56.51** | 66.77 |
| Disent | ✗/✗ | 66.21 | 75.32 | 78.61 | 86.48 | 79.31 | 88.54 | 56.26 | 69.61 |
| + ORBIS | ✗/✗ | 70.40 | 77.63 | **84.22** | 87.82 | **81.00** | **92.53** | 56.43 | **71.13** |

## 5.1 EXPERIMENTAL SETTING

**Baselines and Datasets.** To assess the effectiveness of our proposed laogirtmh in mitigating dataset bias, we conduct a comparative analysis against several existing debiasing algorithms, including Vanilla (without any techniques for debiasing), HEX Wang et al. (2019), LNL Kim et al. (2019), EnD Tartaglione et al. (2021), ReBias Bahng et al. (2020a), LfF Nam et al. (2020), Disent Lee et al. (2021), and BiasEnsemble Lee et al. (2022). In our evaluation, we apply ORBIS to the previous metioned algorithms, specifically Vanilla, LfF, and Disent, while ensuring that the network architecture and hyperparameters remain consistent with their respective experimental settings. By comparing the performance of ORBIS with these algorithms, our goal is to demonstrate its superiority in mitigating dataset bias. We evaluate the performance across various benchmark datasets and provide an in-depth analysis of the results, shedding light on the strengths and advantages of ORBIS.

Our evaluations encompass a divers set of datasets, each presenting unique biases and challenges. These datasetinclude: biased FFHQ (bFFHQ) Kim et al. (2021), biased action recognition (BAR) Nam et al. (2020), Dogans and Cats (DnC) Lee et al. (2022), and CelebA Sagawa et al. (2019). bFFHQ is tailored for age classifiaction and exhibits gender bias, with a majority of "young" samples being female and "old" samples being male. BAR focuses on six different actions and introduces background bias, such as the predominance of "RockWall" backgrounds in the "Climbing" class. DnC showcases color bias, with the majority of images in the "Dogs" class featuring bright-clored dogs, while "Dark" dogs represent bias-conflicting samples. CelebA includes multiple target attributes, and we specifically consider two classification tasks related to gender-biased attributes, namely "Heavymakeup" and "BlondHair." For synthetic dataset (bFFHQ, BAR, and DnC) we evaluate different bias-conflicting ratios, specifically {0.5%, 1%, 2% and 5%} for bFFHQ, and {1% , 5%} for the others.

**Open dataset.** We employ the widely recognized WebVision Li et al. (2017) as $\mathcal{D}_{\text{open}}$ for all of our experiments. The WebVision dataset is obtained through web-crawling and comprises a substantial collection of $980,449$ samples. It is frequently used in research related to noisy labels. It is crucial to emphasize that in our experiments, we do not utilize the labels provided with the WebVision dataset.

**Implementation.** In the implementation of ORBIS, we train the open dataset self-supervised model using SimCLR Chen et al. (2020a). Specifically, we train a ResNet-50 He et al. (2016) model on the WebVision dataset once and utilized it for all benchmarks. Furthermore, we evaluate ResNet-18 trained from scratch. We run 3 epochs for the ResNet-50 case, and $50,000$ iterations for ResNet-18 case. Except for training iterations, we adhered to the training recipes provided in Lee et al. (2022),

including learning rate, weight decay, optimizer, momentum, and learning rate scheduler, which were consistent across all benchmarks. The two primary hyperparameters specific to our approach were set as follows: the similarity threshold $\tau = 0.8$ and balancing parameter $\lambda = 0.01$ for all cases.

## 5.2 PERFORMANCE ANALYSIS

**Synthetically biased benchmarks.** In Table 4, we demonstrate the performance on various benchmarks trained from scratch and fine-tuning the model obtained from SimCLR training. As in the upper side of Table 4, ORBIS improves prior methods and shows performance and denotes the best performance on all cases. In short, LfF + BE + ORBIS shows the best performance on bFFHQ dataset and some DnC and BAR datasets, and Disent + BE + ORBIS shows the best remain settings. Especially, in bFFHQ with 5% case, LfF + BE + ORBIS improves 3.94% point compare to the second best performance, LfF + BE. On the other case, loading SimCLR trained model shows also best performance when we utilize ORBIS. As in the bottom block of Table 4, ORBIS with LfF or Disent shows the best performance for all cases. Therefore, we can conclude that ORBIS can be a good add-on module for the previous debiasing algorithms.

**Real-world benchmark.** CelebA is a real-world benchmark known for the spurious correlation between `Blond hair – Gender` and `Heavy Makeup – Gender`, respectively Sagawa et al. (2019). We use the ResNet-50 model pretrained with the SimCLR algorithm to evaluate the CelebA dataset. We report the average accuracy of the test dataset. As described in Table 5, applying ORBIS to the Vanilla LfF and Disent methods improves perfor-

Table 5: Real-world benchmark, CelebA.

| Algorithm | CelebA | | | |
|---|---|---|---|---|
| | Heavy Makeup | | Blod Hair | |
| | Avg. | Conf. | Avg | Conf. |
| Vanilla | 91.02 | 72.03 | 94.77 | 89.57 |
| + ORBIS | 92.31 | 73.64 | **95.31** | 95.36 |
| LfF | 90.41 | 77.04 | 93.54 | 94.53 |
| + ORBIS | **93.67** | **84.13** | 95.27 | 97.31 |
| Disent | 90.11 | 76.25 | 83.72 | 94.24 |
| + ORBIS | 93.47 | 82.17 | 94.33 | **98.19** |

mance. Specifically, LfF + ORBIS shows the best performance in the cases of `Heavy makeup` and Disent + ORBIS shows best bias-conflicting performance on `Blond hair` targets. Therefore, we can conclude that leveraging open datasets based on ORBIS can improve the debiasing performance on real-world datasets.

## 5.3 ANALYSIS

In this section, we will answer the following questions: (1) we check that which samples are selected by the first step of ORBIS. (2) To verify the impact of each step, we describe ablation study of the proposed algorithm (Appendix D). (3) To check the hyperparameter sensitivity, we examine the performance of various $\tau$ and $\lambda$ (Appendix E).

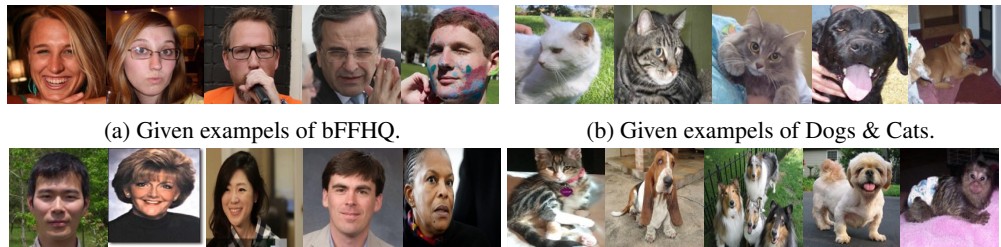

(a) Given exampels of bFFHQ.          (b) Given exampels of Dogs & Cats.

(c) Images in the relevant dataset for bFFHQ.    (d) Image in the relevant dataset for Dogs & Cats.

Figure 3: We plots for two benchmarks, bFFHQ, Dogs & Cats, and the $\mathcal{D}_{\text{rel}}$ sampled from WebVision dataset. At the top row, we describe the bFFHQ and DnC dataset. Two bottom rows describe $\mathcal{D}_{\text{rel}}$.

**Relevant set $\mathcal{D}_{\text{rel}}$ selection.** We analyse what samples are selected at the first stage of ORBIS. Especially, we plot the samples of three benchmarks: bFFHQ, Dogs & Cats. By comparing the examples in the given dataset $\mathcal{D}_{\text{tr}}$, Figure 3a and 3b, and extracted samples Figure 3c and 3d, our first step can extract similar samples to the given training dataset. However, in some cases, similar but may not be in the training dataset, *e.g.,* `Monkey` in the most right image in Figure 3d, can be sampled. Therefore, the proposed method can extract related samples so that it can improve the generalization performance as described in Chen et al. (2020a); Kim et al. (2023a).

## 6 RELATED WORK

We describe related work which is relevant to our work in two perspectives: (1) mitigating dataset bias with or without human supervision. (2) Leveraging open datasets. for each paragraph.

**Dataset bias with or without human supervision.** The most easiest way of utilizing human supervision is to use bias-related labels. For example, Tartaglione et al. (2021); Kim et al. (2019); Singh et al. (2020); Teney et al. (2021); Cadene et al. (2019); Arjovsky et al. (2019) leverage explicit bias labels. On the other hand, the authors of Goyal et al. (2017; 2020) generates labels by using human hands whenever the model requires a specific label related to the bias. However, they requires extensive human labor. Therefore, some works Li & Vasconcelos (2019); Lee et al. (2019); Bahng et al. (2020b) search the way of reducing human involvement by assuming that the model designer knows bias types such as `Gender` is biased in bFFHQ dataset. Another way of reducing human const, leveraging unbiased validation dataset was studied Liu et al. (2021); Nam et al. (2022); Zhang et al. (2022); Lee et al. (2023). They directly utilized validation dataset Nam et al. (2022) or indirectly utilized for hyperparameter tuning. In Hardt et al. (2016); Woodworth et al. (2017); Pleiss et al. (2017); Agarwal et al. (2018), the authors aimed to improve fairness by utilizing given all features.

Despite training unbiased model by using human labor, it requires expensive cost. Therefore, several works aims to minimize human labor. The authors of Kim et al. (2021); Le Bras et al. (2020); Ahn et al. (2023); Darlow et al. (2020) modify the training dataset and make a balanced training dataset. Oppositely, Lee et al. (2021); Nam et al. (2020) proposed weighted training methods which multiply weights to the bias-conflicting samples. The authors of Arjovsky et al. (2019), proposed regularizer to make the trained model have unbiased mind. Because we cannot access the bias-related information, Sohoni et al. (2020); Seo et al. (2022); Creager et al. (2021); Hwang et al. (2022) proposed clustering based approaches, which divide the training dataset into bias-aligned and bias-conflicting samples. Recently, utilizing an ensemble of fully connected layer, called the bias-committee Kim et al. (2022) to detect the bias-conflicting samples was proposed. The authors of Park et al. (2023) proposed network pruning to reduce the impact of bias-aligned samples, and Kim et al. (2023b) leveraged vision-language models to detect bias in the training dataset. On the other side, the authors of Li & Xu (2021); Lang et al. (2021); Krishnakumar et al. (2021) solves the existance of bias in the training dataset, which is called bias identification problem.

**Utilizing open dataset.** Open datasets contain a wealth of divers samples but they are often unlabeled or incorrectly labeled Li et al. (2017). To harness the potential of this valuable resource, several works have proposed strategies for its utilization. In Wei et al. (2021) and Wei et al. (2022), the authors leveraged open datasets to address noisy labeling and class imbalance inssues, respectively. From a self-supervised learning perspective, Kim et al. (2023a) selected and utilized relevant datasets from the open dataset pool. On the other hand, the availability of open datasets is assumed and used in open set recognition Chen et al. (2021); Scheirer et al. (2012); Bendale & Boult (2015); Vaze et al. (2021), webly supervised learning Chen & Gupta (2015); Li et al. (2020), and semi-supervised learning Chen et al. (2020b); Killamsetty et al. (2021); Saito et al. (2021).

## 7 CONCLUSION

In this work, we propose a way of leveraging open datasets to mitigate the dataset bias problem. This is motivated by the fact that when we increase the quantity of bias-conflicting samples in the training dataset, the model can be debiased more easily. To this end, we introduce an algorithm called `ORBIS`, which consists of two major steps. First the proposed algorithm extracts relevant datasets from the abundant irrelevant instances in the open dataset. Afterward, `ORBIS` runs the debiasing method with contrastive loss which is computed based on the given training dataset and obtained relevant samples. Here, the reason why we utilize unsupervised loss is that pseudo-labeling cannot cause performance improvement due to their underline philosohpy. identifies bias-conflicting samples by generating their pseudo-labels. Because the proposed method handles the open dataset, it can be readily applied to previously proposed methods. We further analyze how this open dataset handling method can enhance debiasing performance across various benchmarks. We believe that this open dataset-based approach can serve as a powerful orthogonal research direction for mitigating dataset bias problem.

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

# Appendix
# ORBIS: Open Dataset Can Rescue You from Dataset Bias

Due to the page limitations of the main manuscript, we provide detailed information in this supplementary material as follows: (1) Manually selected samples for experiment in section 3. (2) Pseudo-code of the proposed algorithm Appendix B. (3) Experimental setting details, including learning rate, trained model, and hyperparameters Appendix C. (4) Component analysis, especially the impact of the contrastive loss, open dataset, and relevant cases Appendix D. (5) Hyperparameter sensitivity Appendix E. (6) Pseudo-labeling results on the FixMatch algorithm in section 3 in Appendix F.

## A    MANUALLY SELECTED SAMPLES FOR EXPERIMENT IN SECTION 3

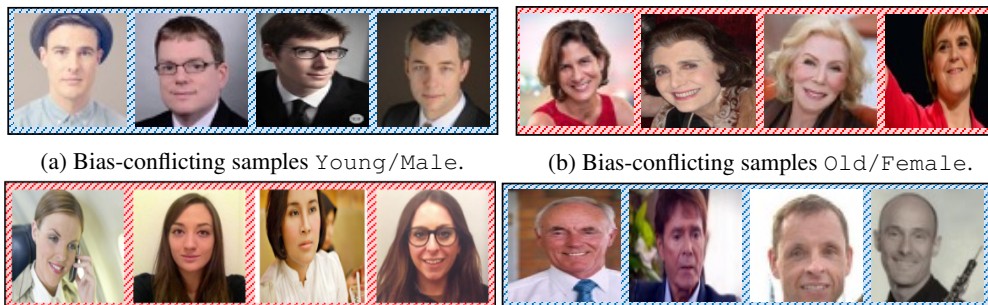

(a) Bias-conflicting samples `Young/Male`.    (b) Bias-conflicting samples `Old/Female`.

(c) Bias-aligned samples `Young/Female`.    (d) Bias-aligned samples `Old/Male`.

Figure 4: Manully selected bias-conflicting/bias-aligned samples in the WebVision dataset for bFFHQ dataset.

To evaluate the impact of relevant samples, we manually selected bias-conflicting samples from the WebVision dataset. As depicted in Figure 4, we provide examples of bias-conflicting and bias-aligned samples selected from the WebVision Li et al. (2017) dataset. Since the bias-conflicting and bias-aligned types of the bFFHQ benchmark are `Young/Mal`, `Old/Female` and `Old/Female` , `Young/Male`, we manually selected 50 samples for each class of bias-conflicting samples and 100 samples for each class of bias-aligned samples.

## B    ALGORITHM DESCRIPTION

In this section, we describe the proposed algorithm in detail. The proposed algorithm is composed of two main steps and preliminary step. In the preliminary step (Step 0), we train the model using the open dataset. However, since we cannot obtain clean and task-related labels, we leverage self-supervised learning, particularly SimCLR Chen et al. (2020a). Subsequently, we extract features from the training dataset $\mathcal{D}_{tr}$ and obtain per-class centroid $C_c$. Based on a predefined hyperparemeter $\tau$, ORBIS selects a highly relevant sample set $\mathcal{D}_{rel}$ from the open dataset $\mathcal{D}_{open}$, where the cosine similarity betwee the per-class centroid and each sample in the open dataset is higher than $\tau$. After obtaining the relevant samples, we ultimately train the debiased model using a debiasing algorithm. In this training, we utilize the contrastive learning loss $\mathcal{L}_{con}$ for the union of $\mathcal{D}_{tr} \cup \mathcal{D}_{rel}$. The entire procedure is described in algorithm 1.

## C    EXPERIMENTAL DETAILS

We utilize two types of networks, ResNet-18 and ResNet-50 for the synthetically biased dataset, and ResNet-50 for the real-world dataset, namely the CelebA dataset. As described in the main document, for the ResNet-18 case, we follow the training recipe exactly as presented in a previous paper Lee et al. (2022). In short we train $50,000$ iterations with a mini-batch size of $64$. The learning rate is $0.0001$ and we apply weight decay with ratio of $0.1$ every $10,000$ steps.

---

**Algorithm 1:** `ORBIS`: Open dataset rescue you from dataset bias

---

**Input:** Training dataset $\mathcal{D}_{\text{tr}}$, Biased feature extractor $f(x)$, Projection head $h(f(x))$, Linear classifier $g(f(x))$.

**Output:** Debiased model $f$ and $g$.

**Initialize:** All biased/debiased models, $f$, $h$, and $g$.

**/* Step 0: Self-supervised Training */**
Train model $f(x)$ based on SimCLR self-supervised learning method.

**/* Step 1: Select Relevant set $\mathcal{D}_{\text{rel}}$ */**
$\mathsf{C}_c = \frac{\sum_{(x,y) \in \mathcal{D}_{\text{tr}}^c} f(x)}{|\mathcal{D}_{\text{tr}}^c|}$    where $\mathcal{D}_{\text{tr}}^c = \{(x,y)|y = c, (x,y) \in \mathcal{D}_{\text{tr}}\}$       `// equation 1`
$\mathcal{D}_{\text{rel}} = \bigcup_{c=1}^{C} \mathcal{D}_{\text{rel}}^c$ where $\mathcal{D}_{\text{rel}}^c = \{x|\mathsf{sim}(\mathsf{C}_c, f(x)(x) \geq \tau \quad \forall x \in \mathcal{D}_{\text{open}}\}$  `// equation 4`
 `and equation 3`

**/* Step 2: Training debised model*/**
**for** *epoch* $< E$ **do**
$\quad$ Compute $\mathcal{L}_{\text{tr}}(g(f(x_i)))$ on $(x_i, y_i) \in \mathcal{D}_{\text{tr}}$
$\quad$ Compute $\mathcal{L}_{\text{con}}(h(f(x_j)))$ on $(x_j) \in \mathcal{D}_{\text{rel+tr}}$                         `// equation 5`
$\quad$ Train $f, h, g$ using $\mathcal{L} = \mathcal{L}_{\text{tser}} + \lambda \mathcal{L}_{\text{con}}$                         `// equation 6`
**end**

---

In the case of ResNet-50, we run training for 10 epochs for the bFFHQ and DnC dataset,s 50 epochs for BAR and 3 epochs for the CelebA dataset. The reason for larger number of epochs for BAR is the extremely smal ldataset size. In the cases of ResNet-50 except for CelebA, the learning rate is 0.0001 and the batch size remains identical to the ResNet-18 Case. We utilize learning rate of 0.00001 with 256 batch size for CelebA dataset.

## D COMPONENT ANALYSIS

Table 6: Component analysis.

| Option | 0.5% | 1% | 2% | 5% |
|---|---|---|---|---|
| Vanilla | 55.64 | 60.96 | 69.00 | 82.88 |
| Only Contrastive | 56.60 | 62.80 | 72.03 | 83.21 |
| Contrastive + Open (Random) | 56.62 | 62.20 | 70.63 | 83.42 |
| Contrastive + Open (Relevant) | **59.23** | **64.43** | **73.17** | **84.32** |

To assess the impact of each component, we evaluate the model by removing individual components. (1) Only Constrative: without leveraging open dataset, we can add contrastivee loss computed by using the given dataset. It can give us information about the impact of using open dataset. (2) Contrative + open (Random): when we exclude the relevant dataset selection, the open dataset can still be utilized to compute the contrastive loss. In this case, we can check whether out-of-distribution samples, *i.e.,* irrelevant samples, are injected into the training procedure. (3) Contrastive + open (relevant): We verify the impact of relevant samples. As described in Table 6, we can conclude that every component is necessary to achieve the best performance.

## E HYPERPARAMETER SENSITIVITY

To assess the sensitivity of the hyperparameters we used, $\tau$ and $\lambda$, we examine the performance across various values of $\tau$ and $\lambda$. We train avanilla model on the bFFHQ dataset with a $\rho = 0.5$ bias-conflicting ratio. As illustrated in Figure 5, `ORBIS` consistently demonstrates superior performance compared to the case without `ORBIS`, represented by the black dotted line. This indicates that `ORBIS` enhaces performance regardless of the values assigned to $\tau$ and $\lambda$.

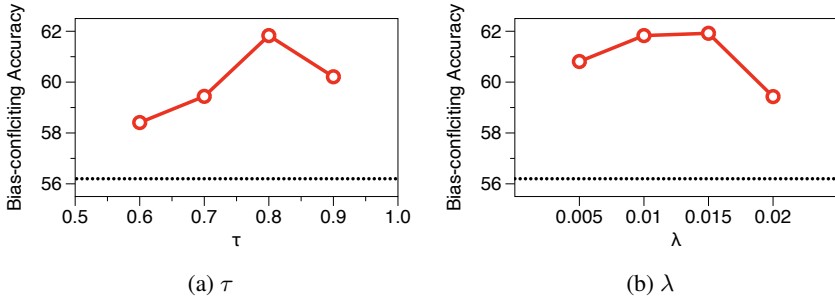

Figure 5: Hyperparameter Sensitivity

(a) $\tau$          (b) $\lambda$

## F   RESULT OF FIXMATCH

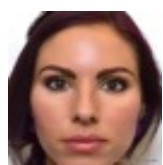 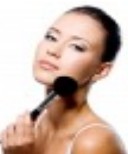 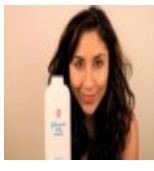 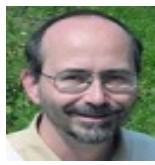 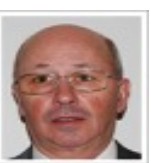 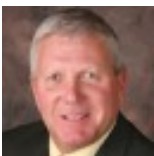

Figure 6: Pseudo-labeling results based on FixMatch.

Fixmatch Sohn et al. (2020) is one of the well-known Semi-Supervised Learning (SSL) algorithms. Like many recent SSL algorithms, FixMatch selects and assigns pseudo-labels to samples with high confidence as a fundamental philosohpy. The detailed training loss for the unlabeled dataset is as follows:

$$\ell_u = \frac{1}{\mu B} \sum_{b=1}^{\mu B} \mathbb{1}(\max(q_b) \geq \tau) H(\hat{q}_b, p_m(y|\mathcal{A}(u_b))).$$

Here, $\tau$ is the threshold hyperparameter, $q_b$ is the output of the weakly augmented sample, $\hat{q}_b$ is the predicted label of $q_b = \arg\max q_b$, and $H$ is the shannon entropy. In simpler terms, samples with high confidence scores, where the maximum softmax value $\max q_b$ is greater than $\tau$, are included in the training procedure.

From this perspective, we plotted the samples with confidence scores higher than $\tau$. As depicted in Figure 6, the sampled images primarily represent `Young/Female` and `Old/Male`. This implieds that the labeled samples are mainly bias-aligned samples, rather than bias-conflicting samples. In other words, pseudo-labeling, especially confidence score-based approaches, struggle to identify bias-conflictign samples. Consequently, these methods may not offer significant benefits from a debiasing standpoint.

