# OpenReview forum: "ORBIS: Open Dataset Can Rescue You From Dataset Bias"
_ICLR.cc/2024/Conference — Submitted to ICLR 2024_

### Official Review · Reviewer_i93n · 2023-10-29

**Soundness:** 2 fair
**Presentation:** 2 fair
**Contribution:** 2 fair
**Rating:** 3
**Confidence:** 4

**Summary:**

The paper focuses on alleviating dataset bias, i.e., spurious correlation between semantic concepts and image characteristics. It proposes to use an external open dataset sampled outside the target domain data to help. It studies selection of relevant samples from such an open dataset. The principle is that sampled images should have aligned contexts with the training set and be "bias conflicting". It trains a debiased model using the training set and selected examples. It conducts experiments on both synthetic and real-world benchmrak datasets.

**Strengths:**

- It makes sense to use data sampled outside the downstream dataset to better solve the downstream task.
- It makes sense to select diverse data relevant to the downstream dataset.

**Weaknesses:**

Below are several concerns related to weaknesses.

- In abstract, it is not clear how to define "open dataset", how to define "noisy information", how to define "characteristics of the given training dataset". Authors should clarify.

- The paper emphasizes the phrase "dataset bias problem" but it is not clear what it means. Is it a new problem proposed in the paper? If so, authors should clarify. Otherwise, authors should cite related papers.

- In Introduction, the paper uses "ski" as a motivational example. Concretely, the paper explains "the images of “ski” exhibit a spurious correlation with “snow,” and the snowy background is considered the bias attribute. Conversely, samples captured in a desert environment (e.g., desert skiing) would be bias conflicting samples." However, isn't it reasonable to have snow in "ski" images. In other words, "ski" is "ski", "desert ski" is a different sport. Authors should clarify.

- Figure 1 is confusing. It is not clear what pink and blue boxes mean, both are labeled as "bias-conflicting".

- In Section 2.1, the examples in the first two paragraphs are conflicting. The writing should be improved.

- Table 2 is hard to read. It has a poor caption.

- In terms of Observation 1, it is confusing to assign arbitrary labels to mined images as a compared method. Shouldn't a reasonable baseline be that of training without extra data? Moreover, what is the dataset used to report numbers for the observations?

- The paper writes "assigns labels based on cosine similarity", it is not clear why using "cosine similarity"? Why not entropy, softmax scores, etc.?

- Table 3 is hard to understand. Does it sample relevant data or just randomly sample data for pseudo labeling? It misses crucial details for understanding.

- The paper motivates the first key point, "avoid focusing solely on selecting bias conflicting samples". But it is not clear why it should "avoid solely selecting bias conflicting samples". Authors should clarify.

- The paper writes "To tackle this issue, we emply a self-supervised learning mechanism that does not rely on target labels". However, images by themselves deliver bias or spurious correlation. So why does self-supervised learning tackle the issue of data bias?

- It is not well motivated to use class centroid in self-supervised learning. Can authors explain this design choice? Why is it better than the typical contrastive loss?

- The debias loss L_{tr} is not defined. Authors should define it before using it.

- The paper writes "Furthermore, we evaluate ResNet-18 trained from scratch". Why not train ResNet50 from scratch?

- There are many typos, e.g., "bset", "for tow reasons", "pseudo-labelign", "a relevant samples", "emply". The paper needs to be polished for good readability.

**Questions:**

Questions are in the weaknesses. I encourage the authors to address them in rebuttal.

**Details Of Ethics Concerns:**

The paper uses face datasets but does not discuss potential ethical issues.

---

### Official Review · Reviewer_1vEM · 2023-11-01

**Soundness:** 2 fair
**Presentation:** 2 fair
**Contribution:** 2 fair
**Rating:** 3
**Confidence:** 5

**Summary:**

The paper proposes using an open dataset to readily mitigate dataset bias with existing debiasing algorithms, called ORBIS. ORBIS consists of two steps. First, it distinguishes the relevant samples from an open dataset, and second, it trains main model with debiasing loss and contrastive loss using these relevant samples. The authors conducted ORBIS with two debiasing algorithms, LfF and Disent.

**Strengths:**

Conducting research on removing dataset bias is beneficial for the entire AI research community.

**Weaknesses:**

W1.The writing quality is very poor.

**Typos**
- Page 1
	- calld -> called
	- adverasarial -> adversarial
- Page 2
	- (Figure 1) Bias-conflicting -> Bias-aligned
	- ...the number of the proportion...-> ...the number or the proportion...
	- bset -> best
- Page 3
	- calssification -> classification
	- v.s. -> vs.
- Page 4
	- consturcting -> constructing
	- implictions -> implications
	- labelign -> labeling
- Page 5
	- (Figure 2) Bias-conflicting -> Bias-aligned
	- anootations -> annotations
	- ...that do note... -> ...that do not...
- Page 6
	- $f$open -> $f_{open}$
	- ...without relying on provided or pseudo-labels. -> without relying on provided [something] or pseudo-labels.
- Page 7
	- laogirtmh -> algorithm
	- datasetinclude -> dataset include
	- Dogans and Cats -> Dogs and Cats
	- classifiaction -> classification
- Page 8
	- examels -> examples
	- plots -> plot
	- three benchmarks -> two benchmarks
- Page 9
	- philosohpy -> philosophy

**Inconsistency between results and explanation.**
- Page 4: 'As indicated in Table 3 semi-supervised debiasing is not a straightforward process In and can potentially degrade performance.'
- The authors described that FixMatch degrades performance, but Table 3 shows that FixMatch didn't degrade performance.

**Lack of information**
- On page 2, in the third contribution point, what type of labels are not required?
- What metric is reported in Table 1?
- On page 4, in Observation 1, the training instructions are missing. Did you combine the relevant samples with the original training set for training, or did you only use relevant samples?
- What metric is reported in Table 3?

**Incorrect sentence**
- Page 5 'As previously noted, the open dataset D_{open} does not inherently contain samples that are directly relevant to the target task.'
- I think the sentence can be revised to 'As previously noted, the open dataset D_{open} might inherently contain samples that are not directly relevant to the target task.'

W2. If the open dataset is biased, and the class-wise centroid is biased toward bias-aligned samples, could ORBIS still be helpful for debiasing if all relevant samples are bias-aligned?

W3. Is there any problem arising from the differences between the open dataset and the target dataset? What should we do if the domains of the open dataset and the target dataset are different?

W4. The applied debiasing algorithms (LfF and Disent) are limited in terms of reweighting methods and may be considered outdated. Can ORBIS effective with sampling-based debiasing method (eg. PGD), contrastive learning-based debiasing methods (ex. CNC, CDvG), mixup-based debiasing methods (ex. selecmix), and logit correction-based debiasing method (ex. LC)?
PGD: Mitigating dataset bias by using per-sample gradient, ICLR 2023
- CNC: Correct-N-Contrast: A Contrastive Approach for Improving Robustness to Spurious Correlations, ICML 2022
- CDvG: Fighting Fire with Fire: Contrastive Debiasing without Bias-free Data via Generative Bias-transformation, ICML 2023
- SelecMix: Debiased Learning by Contradicting-pair Sampling, NeurIPS 2022
- LC: avoiding spurious correlations via logit correction, ICLR 2023

W5. In Table 5, I believe that the extremely high performance of ERM is due to the use of a backbone trained on the open dataset. Therefore, I speculate that the target dataset used here might be very similar to the open dataset. Consequently, it doesn't seem like the ideal target dataset for verifying the effectiveness of ORBIS.

**Questions:**

Q1. Could you clarify that why does ORBIS not outperform BE?
In table 4, in most cases, the BE outperforms ORBIS, even though BE does not use extra data.

Q2. Could you explain in more detail about following sentence?
- Page 4: '(1) Avoid focusing solely on selecting bias conflicting samples: it is not necessary to exclusively focus on selecting bias-conflicting samples in order to increase the ratio of bias-conflicting instances to improve the accuracy of bias-conflicting samples.'

Q3. (minor question) Why construct two types of mini-batches, one is sampled from $D_{tr}$ and the other is sampled from $D_{rel+tr}$? Why are $L_{tr}$ and $L_{con}$ calculated using different samples, even though they are sampled from same dataset $D_{tr}$? Is there any difference in aspect of performance?

---

### Official Review · Reviewer_LB6u · 2023-11-09

**Soundness:** 2 fair
**Presentation:** 1 poor
**Contribution:** 2 fair
**Rating:** 3
**Confidence:** 2

**Summary:**

This paper aims to address dataset bias by using clustering on open datasets to collect additional samples and integrating those with the existing training dataset for training a debiased model. They find that their method can be combined with existing algorithms to outperform previous results on bFFHQ, Dogs & Cats, and BAR.

**Strengths:**

- problem studied is a good extension of existing studies, given the increasing popularity of open and web-scraped datasets
- proposed method is simple and extends to various settings

**Weaknesses:**

1. contains a lot of typos and poor grammatical structure, making the paper hard to read
2. similarly, some tables lack captions and are unclear
3. experiment setup in section 3 is confusing
    - in observation 1, random samples with arbitrarily assigned labels is more like poisoning rather than taking a "natural" sample
    - in observation 2, table and description can be cleaned up to make the conclusion more clear
4. lacks ablation experiments for design design decisions made in section 4
5. observation 2 seems to be key to performance but is not explicitly used in the author's method

**Questions:**

See weaknesses

---

### Meta-Review · Area_Chair_wzU2 · 2023-12-05

**Metareview:**

A) ORBIS proposes a method to mitigate dataset bias using open dataset
B) Dataset bias is an important problem that needs to be tackled
C) Reviewers agree that the paper appears to be poorly written, has inconsistent results, lack of detail in important parts.

**Justification For Why Not Higher Score:**

Reviewers agree paper is poorly written.

**Justification For Why Not Lower Score:**

N/A

---

### Decision · Program_Chairs · 2024-01-16

Reject